# Fair Allocation of Indivisible Chores: Beyond Additive Costs

**Bo Li**[1]        **Fangxiao Wang**[1]        **Yu Zhou**[1]

[1]Department of Computing, The Hong Kong Polytechnic University, Hong Kong, China

`comp-bo.li@polyu.edu.hk,`
`fangxiao.wang@connect.polyu.hk, csyzhou@comp.polyu.edu.hk`

## Abstract

We study the maximin share (MMS) fair allocation of $m$ indivisible chores to $n$ agents who have costs for completing the assigned chores. It is known that exact MMS fairness cannot be guaranteed, and so far the best-known approximation for additive cost functions is $\frac{13}{11}$ by Huang and Segal-Halevi [EC, 2023]; however, beyond additivity, very little is known. In this work, we first prove that no algorithm can ensure better than $\min\{n, \frac{\log m}{\log \log m}\}$-approximation if the cost functions are submodular. This result also shows a sharp contrast with the allocation of goods where constant approximations exist as shown by Barman and Krishnamurthy [TEAC, 2020] and Ghodsi et al. [AIJ, 2022]. We then prove that for subadditive costs, there always exists an allocation that is $\min\{n, \lceil \log m \rceil\}$-approximation, and thus the approximation ratio is asymptotically tight. Besides multiplicative approximation, we also consider the ordinal relaxation, 1-out-of-$d$ MMS, which was recently proposed by Hosseini et al. [JAIR and AAMAS, 2022]. Our impossibility result implies that for any $d \geq 2$, a 1-out-of-$d$ MMS allocation may not exist. Due to these hardness results for general subadditive costs, we turn to studying two specific subadditive costs, namely, bin packing and job scheduling. For both settings, we show that constant approximate allocations exist for both multiplicative and ordinal relaxations of MMS.

## 1  Introduction

### 1.1  Background and related research

Although fair resource allocation has been widely studied in the past decade, the research is centered around additive functions [3]. However, in many real-world scenarios, the functions are more complicated. Particularly, the functions appear in machine learning and artificial intelligence (e.g, clustering [37], Sketches [18], Coresets [36], data distillation [41]) are often submodular, and we refer the readers to a recent survey that reviews some major problems within machine learning that have been touched by submodularity [12]. Therefore, in this work, we study the fair allocation problem when the agents have beyond additive cost functions. The mainly studied solution concept is maximin share (MMS) fairness [14], which is traditionally defined for the allocation of goods as a relaxation of proportionality (PROP). A PROP allocation requires that the utility of each agent is no smaller than the average utility when all items are allocated to her. PROP is too demanding in the sense that such an allocation does not exist even when there is a single item and two agents. Due to this strong impossibility result, the maximin share (MMS) of an agent is proposed to relax the average utility by the maximum utility the agent can guarantee herself if she is to partition the items into $n$ bundles but is to receive the least favorite bundle.

37th Conference on Neural Information Processing Systems (NeurIPS 2023).

For the allocation of goods, although MMS significantly weakens the fairness requirement, it was first shown by Kurokawa et al. [31, 32] that there are instances where no allocation is MMS fair for all agents. Accordingly, designing (efficient) algorithms to compute approximately MMS fair allocations steps into the center of the field of algorithmic fair allocation. Kurokawa et al. [32] first proved that there exists a $\frac{2}{3}$-approximate MMS fair allocation for additive utilities, and then Amanatidis et al. [1] designed a polynomial time algorithm with the same approximation guarantee. Later, Ghodsi et al. [23] improved the approximation ratio to $\frac{3}{4}$, and Garg and Taki [22] further improved it to $\frac{3}{4} + o(1)$. On the negative side, Feige et al. [20] proved that no algorithm can ensure better than $\frac{39}{40}$ approximation. Beyond additive utilities, Barman and Krishnamurthy [6] initiated the study of approximately MMS fair allocation with submodular utilities, and proved that a $0.21$-approximate MMS fair allocation can be computed by the round-robin algorithm. Ghodsi et al. [24] improved the approximation ratio to $\frac{1}{3}$, and moreover, they gave constant and logarithmic approximation guarantees for XOS and subadditive utilities, respectively. The approximations for XOS and subadditive utilities are recently improved by Seddighin and Seddighin [40]. There are also several works that delve into concrete combinatorial problems and seek to improve approximation ratios compared to more general functions. Li et al. [33] and Hummel and Hetland [30] introduced interval scheduling and independent set structures, respectively, into MMS fair allocation problems. In both cases, the induced utility functions correspond to special cases of XoS functions. Both of these two works enhanced the approximation ratios for their specific utility functions. Some other combinatorial problems that have been studied for goods include the knapsack problem [21, 9, 8] and matroid constraints [19].

For the parallel problem of chores, where agents need to spend costs on completing the assigned chores, less effort has been devoted. Aziz et al. [4] first proved that the round-robin algorithm ensures 2-approximation for additive costs. Barman and Krishnamurthy [6], Huang and Lu [28] and Huang and Segal-Halevi [29] respectively improved the approximation ratio to $\frac{4}{3}$, $\frac{11}{9}$ and $\frac{13}{11}$. Recently, Feige et al. [20] proved that with additive costs, no algorithm can be better than $\frac{44}{43}$-approximate. However, except a recent work that studies binary supermodular costs [10], very little is known beyond additivity.

Besides multiplicative approximations, we also consider the ordinal approximation of MMS, namely, 1-out-of-$d$ MMS fairness which is recently studied in [5, 26, 27], and proportionality up to any item (PROPX) which is an alternative way to relax proportionality [38, 34]. For indivisible chores, Hosseini et al. [26] and Li et al. [34] respectively proved the existence of 1-out-of-$\lceil \frac{3n}{4} \rceil$ MMS fair and exact PROPX allocations. All the above works also assume additive costs. More works related to this paper in the literature can be seen in Appendix A.

## 1.2 Main results

In this work, we aim at understanding the extent to which MMS fairness can be satisfied when the cost functions are beyond additive. In a sharp contrast to the allocation of goods, we first show that no algorithm can ensure better than $\min\{n, \frac{\log m}{\log \log m}\}$-approximation when the cost functions are submodular.[1] Further, we show that for general subadditive cost functions, there always exists an allocation that is $\min\{n, \log m\}$-approximate MMS, and thus the approximation ratio is asymptotically tight. Next, we consider the ordinal relaxation, 1-out-of-$d$ MMS. It is trivial that 1-out-of-1 MMS is satisfied no matter how the items are allocated, and somewhat surprisingly, our impossibility result implies that for any $d \geq 2$, there is an instance for which no allocation is 1-out-of-$d$ MMS.

**Result 1** For general subadditive cost functions, the asymptotically tight multiplicative approximation ratio of MMS is $\min\{n, \log m\}$. Further, for any $d \geq 2$, a 1-out-of-$d$ MMS allocation may not exist.

Result 1 combines Theorems 1, 2 and Corollary 1. The strong impossibility in Result 1 does not rule out the possibility of constant multiplicative or ordinal approximation of MMS fair allocations for specific subadditive costs. For this reason, we turn to studying two concrete settings with subadditive costs. The first setting encodes a bin packing problem, which has applications in many areas (e.g., semiconductor chip design, loading vehicles with weight capacity limits, and filling containers [17]). In the first setting, the items have sizes which can be different to different agents. The agents have bins that can be used to pack the items allocated to them with the goal of using as few bins as possible. The second setting encodes a job scheduling problem, which appears in many research

---

[1]In this paper we use $\log(\cdot)$ to denote $\log_2(\cdot)$.

areas, including data science, big data, high-performance computing, and cloud computing [25]. In the second setting, the items are jobs that need to be processed by the agents. Each job may be of different lengths to different agents and each agent controls a set of machines with possibly different speeds. Upon receiving a set of jobs, an agent's cost is determined by the corresponding minimum completion time when processing the jobs using her own machines (i.e., makespan). As will be clear, job scheduling setting is more general than the additive cost setting. Besides, it uncovers new research directions for group-wise fairness.

**Result 2** For the bin packing and job scheduling settings, a 1-out-of-$\lfloor \frac{n}{2} \rfloor$ MMS allocation and a 2-approximate MMS allocation always exist.

Result 2 combines Theorems 3, 4 and Corollaries 2, 3. Besides studying MMS fairness, in Appendix F, we also prove hardness results for two other relaxations of proportionality, i.e., PROP1 and PROPX.

## 2   Preliminaries

For any integer $k \geq 1$, let $[k] = \{1, \ldots, k\}$. In a fair allocation instance $I = (N, M, \{v_i\}_{i \in N})$, there are $n$ agents denoted by $N = [n]$ and $m$ items denoted by $M = \{e_1, \ldots, e_m\}$. Each agent $i \in N$ has a cost function over the items, $v_i : 2^M \to \mathbb{R}^+ \cup \{0\}$. Note that for simplicity, we abuse $v_i(\cdot)$ to denote a cost function. The items are chores, and particularly, upon receiving a set of items $S \subseteq M$, $v_i(S)$ represents the effort or cost agent $i$ needs to spend on completing the chores in $S$. The cost functions are normalized and monotone, i.e., $v_i(\emptyset) = 0$ and $v_i(S_1) \leq v_i(S_2)$ for any $S_1 \subseteq S_2 \subseteq M$. Note that no bounded approximation can be achieved for general cost functions, and we provide one such example in Appendix B.1. Thus we restrict our attention to the following three classes. A cost function $v_i$ is subadditive if for any $S_1, S_2 \subseteq M$, $v_i(S_1 \cup S_2) \leq v_i(S_1) + v_i(S_2)$. It is submodular if for any $S_1 \subseteq S_2 \subseteq M$ and any $e \in M \setminus S_2$, $v_i(S_2 \cup \{e\}) - v_i(S_2) \leq v_i(S_1 \cup \{e\}) - v_i(S_1)$. It is additive if for any $S \subseteq M$, $v_i(S) = \sum_{e \in S} v_i(\{e\})$. It is widely known that any additive function is also submodular, and any submodular function is also subadditive.

An allocation $\mathbf{A} = (A_1, \ldots, A_n)$ is an $n$-partition of the items where $A_i$ contains the items allocated to agent $i$ such that $A_i \cap A_j = \emptyset$ for any $i \neq j$ and $\bigcup_{i \in N} A_i = M$. For any set $S$ and integer $d$, let $\Pi_d(S)$ be the set of all $d$-partitions of $S$. The maximin share (MMS) of agent $i$ is

$$\mathsf{MMS}_i^n(I) = \min_{(X_1, \ldots, X_n) \in \Pi_n(M)} \max_{j \in [n]} v_i(X_j).$$

Note that we may neglect $n$ and $I$ in $\mathsf{MMS}_i^n(I)$ when there is no ambiguity. Note that the computation of MMS is NP-hard even when the costs are additive, which can be verified by a reduction from the Partition problem. Given an $n$-partition of $M$, $\mathbf{X} = (X_1, \ldots, X_n)$, if $v_i(X_j) \leq \mathsf{MMS}_i$ for any $j \in [n]$, then $\mathbf{X}$ is called an *MMS defining partition* for agent $i$. Note that the original definition of $\mathsf{MMS}_i$ for chores is defined with non-positive values, where the minimum value of the bundles is maximized. In this work, to simplify the notions, we choose to use non-negative numbers (representing costs), and thus the definition is equivalently changed to be the maximum cost of the bundles being minimized. To be consistent with the literature, we still call it maximin share.

**Definition 1 ($\alpha$-MMS)** *An allocation $\mathbf{A} = (A_1, \ldots, A_n)$ is $\alpha$-approximate maximin share ($\alpha$-MMS) fair if $v_i(A_i) \leq \alpha \cdot \mathsf{MMS}_i$ for all $i \in N$. The allocation is MMS fair if $\alpha = 1$.*

Given the definition of MMS, for any agent $i$ with subadditive cost $v_i(\cdot)$, we have the following bounds for $\mathsf{MMS}_i$,

$$\mathsf{MMS}_i \geq \max \Big\{ \max_{e \in M} v_i(\{e\}), \frac{1}{n} \cdot v_i(M) \Big\}. \tag{1}$$

Following recent works [5, 27, 26], we also consider the ordinal approximation of MMS, namely, 1-out-of-$d$ MMS fairness. Intuitively, MMS fairness can be regarded as 1-out-of-$n$ MMS (i.e., partitioning the items into $n$ bundles but receiving the largest bundle). Since 1-out-of-$n$ MMS allocations may not exist, we can instead find a maximum integer $d \leq n$ such that a 1-out-of-$d$ MMS allocation is guaranteed to exist. Given a $d$-partition of $M$, $\mathbf{X} = (X_1, \ldots, X_d)$, if $v_i(X_j) \leq \mathsf{MMS}_i^d$ for any $j \in [d]$, then $\mathbf{X}$ is called a *1-out-of-$d$ MMS defining partition* for agent $i$. An allocation $\mathbf{A}$ is *1-out-of-$d$ MMS fair* if for every agent $i \in N$, $v_i(A_i) \leq \mathsf{MMS}_i^d$. More generally, given any $\alpha \geq 1$, we have the bi-factor approximation, $\alpha$-approximate 1-out-of-$d$ MMS, if $v_i(A_i) \leq \alpha \cdot \mathsf{MMS}_i^d$ for

every $i \in N$. By the definition, we have the following simple observation, whose proof is deferred to Appendix B.2.

**Observation 1** *For any instance with subadditive costs, given any integer $1 \leq d \leq n$, a 1-out-of-$d$ MMS allocation is $\lceil \frac{n}{d} \rceil$-MMS fair.*

## 3 General subadditive cost setting

By Inequality 1, if the costs are subadditive, allocating all items to a single agent ensures an approximation of $n$, which is the most unfair algorithm. Surprisingly, such an unfair algorithm achieves the optimal approximation ratio of MMS even if the costs are submodular.

**Theorem 1** *For any $n \geq 2$, there is an instance with submodular costs for which no allocation is better than $n$-MMS or $\frac{\log m}{\log \log m}$-MMS.*

**Proof.** For any fixed $n \geq 2$, we construct an instance that contains $n$ agents and $m = n^n$ items. By taking logarithm of $m = n^n$ twice, it is easy to obtain

$$n = \frac{\log m}{\log \log m - \log \log n} \geq \frac{\log m}{\log \log m}.$$

Thus, in the following, it suffices to show that no allocation can be better than $n$-MMS. Let each item correspond to a point in an $n$-dimensional coordinate system, i.e.,

$$M = \{(x_1, x_2, \ldots, x_n) \mid x_i \in [n] \text{ for all } i \in [n]\}.$$

For each agent $i \in N$, we define $n$ covering planes $\{C_{il}\}_{l \in [n]}$ and for each $l \in [n]$,

$$C_{il} = \{(x_1, x_2, \ldots, x_n) \mid x_i = l \text{ and } x_j \in [n] \text{ for all } j \in [n] \setminus \{i\}\}. \tag{2}$$

Note that $\{C_{il}\}_{l \in [n]}$ forms an exact cover of the points in $M$, i.e., $\bigcup_l C_{il} = M$ and $C_{il} \cap C_{iz} = \emptyset$ for all $l \neq z$. For any set of items $S \subseteq M$, $v_i(S)$ equals the minimum number of planes in $\{C_{il}\}_{l \in [n]}$ that can cover $S$. Therefore, $v_i(S) \in [n]$ for all $S$. We first show $v_i(\cdot)$ is submodular for every $i$. For any $S \subseteq T \subseteq M$ and any $e \in M \setminus T$, if $e$ is not in the same covering plane as any point in $T$, $e$ is not in the same covering plane as any point in $S$, either. Thus, $v_i(T \cup \{e\}) - v_i(T) = 1$ implies $v_i(S \cup \{e\}) - v_i(S) = 1$, and accordingly,

$$v_i(T \cup \{e\}) - v_i(T) \leq v_i(S \cup \{e\}) - v_i(S).$$

Since $\{C_{il}\}_{l \in [n]}$ is an exact cover of $M$, $\mathsf{MMS}_i = 1$ for every $i$, where the MMS defining partition is simply $\{C_{il}\}_{l \in [n]}$. Then to prove the theorem, it suffices to show that for any allocation of $M$, there is at least one agent whose cost is $n$. For the sake of contradiction, we assume there is an allocation $\mathbf{A} = (A_1, \ldots, A_n)$ where every agent has cost at most $n - 1$. This means that for every $i \in N$, there exists a plane $C_{il_i}$ such that $A_i \cap C_{il_i} = \emptyset$. Consider the point $\mathbf{b} = (l_1, \ldots, l_n)$, it is clear that $\mathbf{b} \in C_{il_i}$ and thus $\mathbf{b} \notin A_i$ for all $i$. This means that $\mathbf{b}$ is not allocated to any agent, a contradiction. Therefore, such an allocation $\mathbf{A}$ does not exist which completes the proof of the theorem. $\blacksquare$

To facilitate the understanding of Theorem 1, in Appendix C.1, we visualize an instance with 3 agents and 27 items where no allocation is better than 3-MMS. The hard instance in Theorem 1 also implies the following lower bound for 1-out-of-$d$ MMS, whose proof is in Appendix C.2.

**Corollary 1** *For any $2 \leq d \leq n$, there is an instance with submodular cost functions for which no allocation is 1-out-of-$d$ MMS.*

**Theorem 2** *For any instance with subadditive cost functions, there always exists a $\min\{n, \lceil \log m \rceil\}$-MMS allocation.*

**Proof.** We describe the algorithm that computes a $\min\{n, \lceil \log m \rceil\}$-MMS allocation in Algorithm 1. First, if $\log m \geq n$, we are safe to arbitrarily allocate the items to the agents, which ensures $n$-approximation.

The tricky case is when $\log m < n$, where we cannot allocate too many items to a single agent. For this case, we first look at agent 1's MMS defining partition $\mathbf{D}^1 = (D_1^1, \ldots, D_n^1)$, where $v_1(D_j^1) \leq \mathsf{MMS}_1$

---
**Algorithm 1** Computing a $\min\{n, \lceil \log m \rceil\}$-MMS allocation for subadditive costs
---
**Input:** An instance $(N, M, \{v_i\}_{i \in N})$ with general subadditive costs.
**Output:** An allocation $\mathbf{A} = (A_1, \ldots, A_n)$ such that $v_i(A_i) \leq \lceil \log m \rceil \cdot \mathsf{MMS}_i$ for all $i \in N$.
 1: Initialize $A_i \leftarrow \emptyset$ for every $i \in N$.
 2: **if** $\log m \geq n$ **then**
 3:     $A_1 \leftarrow M$.
 4: **else**
 5:     $i \leftarrow 1$ and $M_0 \leftarrow M$.
 6: **end if**
 7: **while** $M_{i-1} \neq \emptyset$ and $i \leq n$ **do**
 8:     Let $(D_1^i, \ldots, D_n^i)$ be one of $i$'s MMS defining partitions over $M$.
 9:     Let $R_j^i = D_j^i \cap M_{i-1}$ for all $j \in [n]$. Re-index the bundles such that $|R_1^i| \geq \cdots \geq |R_n^i|$.
10:     $A_i \leftarrow \bigcup_{j \in [\lceil \log m \rceil]} R_j^i$ and $M_i \leftarrow M_{i-1} \setminus A_i$.
11:     $i \leftarrow i + 1$.
12: **end while**
---

for all $j \in [n]$ and we assume that they are ordered by the number of items, i.e., $|D_1^1| \geq \cdots \geq |D_n^1|$. In order to ensure that agent 1's cost is no more than $\lceil \log m \rceil$ times her MMS, we ask her to take away $\lceil \log m \rceil$ largest bundles (in terms of number of items) in $\mathbf{D}^1$, i.e., $A_1 = \bigcup_{j \in [\lceil \log m \rceil]} D_j^1$. Since the cost function is subadditive,

$$v_1(A_1) \leq \sum_{j \in [\lceil \log m \rceil]} v_1(D_j^1) \leq \lceil \log m \rceil \cdot \mathsf{MMS}_1.$$

Moreover, since on average each bundle in $\mathbf{D}^1$ contains $\frac{m}{n}$ items and $A_1$ contains the bundles with largest number of items, $|A_1| \geq \lceil \log m \rceil \cdot \frac{m}{n} \geq \frac{\log m}{n} \cdot m$. That is, at least $\frac{\log m}{n}$ fraction of the items are taken away by agent 1. Let $M_1 = M \setminus A_1$ be the set of remaining items, and we have

$$|M_1| \leq \left(1 - \frac{\log m}{n}\right) \cdot m.$$

We next ask agent 2 to take away items in a similar way to agent 1. Let $\mathbf{D}^2 = (D_1^2, \ldots, D_n^2)$ be one of agent 2's MMS defining partitions, and $\mathbf{R}^2 = (R_1^2, \ldots, R_n^2)$ be the remaining items in these bundles, i.e., $R_j^2 = D_j^2 \cap M_1$. Again, we assume $|R_1^2| \geq \cdots \geq |R_n^2|$. Letting $A_2 = \bigcup_{j \in [\lceil \log m \rceil]} R_j^2$ and $M_2 = M_1 \setminus A_2$, we have $v_2(A_2) \leq \lceil \log m \rceil \cdot \mathsf{MMS}_2$. Moreover, since on average each bundle in $\mathbf{R}^2$ contains $\frac{|M_1|}{n}$ items and $A_2$ contains the bundles with largest number of items,

$$|A_2| \geq \lceil \log m \rceil \cdot \frac{|M_1|}{n} \geq \frac{\log m}{n} \cdot |M_1|,$$

which gives

$$|M_2| \leq \left(1 - \frac{\log m}{n}\right) \cdot |M_1| \leq \left(1 - \frac{\log m}{n}\right)^2 \cdot m. \tag{3}$$

We continue with the above procedure for agents $i = 3, \ldots, n$ with the formal description shown in Algorithm 1. It is straightforward that every agent $i$ who gets a bundle $A_i$ has cost at most $\lceil \log m \rceil \cdot \mathsf{MMS}_i$. Further, by induction, Equation 3 holds for all agents $i \leq n$, i.e.,

$$|M_i| \leq \left(1 - \frac{\log m}{n}\right) \cdot |M_{i-1}| \leq \left(1 - \frac{\log m}{n}\right)^i \cdot m.$$

To show the validity of the Algorithm, it remains to show that the algorithm can allocate all items, i.e., $M_n = \emptyset$. This can be seen from the following inequalities,

$$|M_n| \leq \left(1 - \frac{\log m}{n}\right)^n \cdot m = \left(1 - \frac{\log m}{n}\right)^{\frac{n}{\log m} \cdot \log m} \cdot m < \left(\frac{1}{e}\right)^{\log m} \cdot m < \frac{1}{m} \cdot m = 1.$$

Since $|M_n| < 1$, $M_n$ must be empty, which completes the proof of the theorem. ∎

Note that Theorem 1 does not rule out the possibility of beating the approximation ratio for specific subadditive costs. In the next two sections, we turn to studying two specific settings, where we are able to beat the lower bounds in Theorem 1 and Corollary 1 by designing algorithms that can guarantee constant ordinal and multiplicative approximations of MMS. We will mostly consider the ordinal approximation of MMS. By Observation 1, the ordinal approximation gives a result of the multiplicative one, which can be improved by slightly modifying the designed algorithms.

# 4 Bin packing setting

## 4.1 Model

The first setting encodes a bin packing problem where the items have sizes and need to be packed into bins by the agents. The items may be of different sizes to different agents. Specifically, each item $e_j \in M$ has size $s_{i,j} \geq 0$ to each agent $i \in N$. For a set of items $S$, $s_i(S) = \sum_{e_j \in S} s_{i,j}$. Each agent $i \in N$ has unlimited number of bins with the same capacity $c_i$. Without loss of generality, we assume that $c_1 \geq \cdots \geq c_n$ and $c_i \geq \max_{e_j \in M} s_{i,j}$ for all $i \in N$.

Upon receiving a set of items $S \subseteq M$, agent $i$'s cost $v_i(S)$[2] is determined by the minimum number of bins (with capacity $c_i$) that can pack all items in $S$. Note that the calculation of $v_i(S)$ involves solving a classic bin packing problem which is NP-hard. For any two sets $S_1$ and $S_2$, $v_i(S_1 \cup S_2) \leq v_i(S_1) + v_i(S_2)$ since the optimal packing of $S_1 \cup S_2$ is no worse than packing $S_1$ and $S_2$ separately and thus $v_i(\cdot)$ is subadditive. Accordingly, $\mathsf{MMS}_i^d$ is essentially the minimum number $k_i$ such that the items can be partitioned into $d$ bundles and the items in each bundle can be packed into no more than $k_i$ bins. The definition of $\mathsf{MMS}_i^d$ gives $\mathsf{MMS}_i^d \cdot c_i \geq \frac{s_i(M)}{d}$ for all $i \in N$.

We say an item $e_j \in M$ is *large* for an agent $i$ if the size of $e_j$ to $i$ exceeds half of the capacity of $i$'s bins, i.e., $s_{i,j} > \frac{c_i}{2}$; otherwise, we say $e_j$ is *small* for $i$. Let $H_i$ denote the set of $i$'s large items in $M$, and $L_i$ denote the set of $i$'s small items; that is, $H_i = \{e_j \in M : s_{i,j} > \frac{c_i}{2}\}$ and $L_i = \{e_j \in M : s_{i,j} \leq \frac{c_i}{2}\}$. Since two large items cannot be put together into the same bin, the number of each agent $i$'s large items is at most $\mathsf{MMS}_i^d \cdot d$; that is, $|H_i| \leq \mathsf{MMS}_i^d \cdot d$.

We apply a widely-used reduction [13, 28] to restrict our attention on identical ordering (IDO) instances where $s_{i,1} \geq \cdots \geq s_{i,m}$ for all $i$. Specifically, it means that any algorithm that ensures $\alpha$-approximate 1-out-of-$d$ MMS allocations for IDO instances can be converted to compute $\alpha$-approximate 1-out-of-$d$ MMS allocations for general instances. The reduction may not work for all subadditive costs, but we prove in Appendix D.1 that it does work for the bin packing and job scheduling settings. Therefore, for these two settings, we only consider IDO instances.

## 4.2 Algorithm

Next, we elaborate on the algorithm that proves Theorem 3.

**Theorem 3** *A 1-out-of-$\lfloor \frac{n}{2} \rfloor$ MMS allocation always exists for any bin packing instance.*

Let $d = \lfloor \frac{n}{2} \rfloor$. In a nutshell, our algorithm consists of two parts: in the first part, we partition the items into $d$ bundles in a bag-filling fashion and select one or two agents for each bundle. In the second part, for each of the $d$ bundles and each of the agents selected for it, we present an imaginary assignment of the items in the bundle to the bins of the agent. These imaginary assignments are used to guide the allocation of the items to the agents, such that each agent receives cost no more than her 1-out-of-$d$ MMS.

### 4.2.1 Part 1: partitioning the items into $d$ bundles

The algorithm in the first part is formally presented in Algorithm 2, which runs in $d$ rounds of bag initialization (Steps 5 to 8) and bag filling (Steps 12 to 18). For each round $j \in [d]$, we define *candidate agents* - those who think the size of the bag $B_j$ is not large enough and have unallocated small items (Step 4). Note that the set of candidate agents changes with the items in the bag and the

---

[2]Note that although the value of $v_i(S)$ also depends on $c_i$, to simplify the notations, we let the subscript $i$ absorb $c_i$ and neglect an extra parameter in $v_i(\cdot)$.

unallocated items. In the bag initialization procedure, we put into the bag the item $e_j$ and the items every $d$ items after $e_j$ (i.e., $e_{j+d}, e_{j+2d}, \ldots$), as long as they have not been allocated and are large for at least one remaining agent. We select one such agent. After the bag initialization procedure, if there is at most one candidate agent, the round ends and the candidate agent (if exists and has not been selected) is added as another selected agent. Otherwise, we enter the bag filling procedure.

In the bag filling procedure, as long as there exist at least two candidate agents, we let two of them be the selected agents and put the smallest unallocated item into the bag. If there is at most one candidate agent after the smallest item is put into the bag, the round ends and the only candidate agent (if exists and has not been selected) replaces one of the selected agents.

---

**Algorithm 2** Partitioning the items into $d$ bundles.

---

**Input:** An IDO bin packing instance $(N, M, \{v_i\}_{i \in N}, \{s_i\}_{i \in N})$.
**Output:** A $d$-partition of the items $\mathbf{B} = (B_1, \ldots, B_d)$ and disjoint sets of selected agents $\mathbf{G} = (G_1, \ldots, G_d)$.

1: Initialize $L_i \leftarrow \{e_j \in M : s_{i,j} \leq \frac{c_i}{2}\}$ for each $i \in N$, and $R \leftarrow M$.
2: **for** $j = 1$ to $d$ **do**
3:    Initialize $B_j \leftarrow \emptyset$, $G_j \leftarrow \emptyset$, $t \leftarrow j$.
4:    $N(B_j) \leftarrow \{i \in N : s_i(B_j) \leq \frac{s_i(M)}{d}$ and $L_i \cap R \neq \emptyset\}$. // Candidate agents
5:    **while** $e_t \in R$ and there exists an agent $i \in N$ who thinks $e_t$ is large **do**
6:       $B_j \leftarrow B_j \cup \{e_t\}$, $R \leftarrow R \setminus \{e_t\}$, $t \leftarrow t + d$.
7:       $G_j \leftarrow \{i\}$.
8:    **end while**
9:    **if** $|N(B_j)| = 1$ and $N(B_j) \neq G_j$ **then**
10:       Pick $i \in N(B_j)$, $G_j \leftarrow G_j \cup \{i\}$.
11:   **end if**
12:   **while** $|N(B_j)| \geq 2$ **do**
13:       Pick $i_1, i_2 \in N(B_j)$, $G_j \leftarrow \{i_1, i_2\}$.
14:       Pick the smallest item $e \in R$, $B_j \leftarrow B_j \cup \{e\}$, $R \leftarrow R \setminus \{e\}$.
15:       **if** $|N(B_j)| = 1$ and $N(B_j) \not\subseteq G_j$ **then**
16:          Pick $i \in N(B_j)$ and replace one arbitrary agent in $G_j$ with agent $i$.
17:       **end if**
18:   **end while**
19:   $N \leftarrow N \setminus G_j$.
20: **end for**

---

The way we establish the bag and select the agents makes the following two important properties satisfied for every round $j \in [d]$.

- **Property 1**: for each selected agent $i \in G_j$, there are at most $\mathsf{MMS}_i^d$ items in $B_j$ that are large for $i$. Besides, letting $e_j^*$ be the item lastly added to $B_j$, if $e_j^*$ is small for $i$, then
$$s_i(B_j \setminus \{e_j^*\}) \leq \frac{s_i(M)}{d}.$$

- **Property 2**: for each remaining agent $i'$ (i.e., $i' \notin \bigcup_{l \in [j]} G_l$), either $s_{i'}(B_j) > \frac{s_{i'}(M)}{d}$ or no unallocated item is small for $i$ at the end of round $j$. Besides, no item in $\{e_j, e_{j+d}, \ldots\}$ that is large for $i'$ remains unallocated at the end of round $j$.

**Proof.** For the first property, observe that large items are added into the bag only in the bag initialization procedure, where one out of every $d$ items is picked. Since there are at most $d \cdot \mathsf{MMS}_i^d$ large items for every agent $i$, the bag contains at most $\mathsf{MMS}_i^d$ of $i$'s large items. There are two cases where $e_j^*$ is small for an agent $i \in G_j$. First, $i$ is the only candidate agent after $e_j^*$ is added, for which case, we have $s_i(B_j) \leq \frac{s_i(M)}{d}$. Second, $i$ is one of the two selected candidate agents before $e_j^*$ is added, for which case, we have $s_i(B_j \setminus \{e_j^*\}) \leq \frac{s_i(M)}{d}$. In both cases, $s_i(B_j \setminus \{e_j^*\}) \leq \frac{s_i(M)}{d}$ holds. The second property is quite direct by the algorithm, since there is no candidate agent outside $G_j$ at the end of round $j$ (i.e., $N(B_j) \setminus G_j = \emptyset$), and all unallocated large items in $\{e_j, e_{j+d}, \ldots\}$ are put into the bag in the bag initialization procedure. ∎

Property 1 ensures that the items in each bundle $B_j \in \mathbf{B}$ can be allocated to the selected agents in $G_j$, such that each agent $i \in G_j$ can use no more than $\mathsf{MMS}_i^d$ bins to pack all the items allocated to her, which will be shown in the following part. Property 2 ensures the following claim.

**Claim 1** *All the items can be allocated in Algorithm 2.*

**Proof.** Observe that when the last round begins, there are at least $n - (d-1) \cdot 2 \geq 2$ remaining agents. If all the unallocated items are large for some remaining agent, all of them are added into the bag during the bag initialization procedure of the last round and thus no item remains unallocated. Now consider the case where some unallocated item is small for any remaining agent. By Property 2, for any $j \in [d-1]$ and any remaining agent $i'$, we have $s_{i'}(B_j) > \frac{s_{i'}(M)}{d}$. This gives that the total size of the unallocated items to $i'$ is smaller than $\frac{s_{i'}(M)}{d}$. Besides, after the bag initialization procedure of the last round, no large item remains and every remaining item is small for any remaining agent. Combining these two facts, we know that there are always at least 2 candidate agents and thus all small items can be allocated in the bag filling procedure, which completes the proof. ∎

#### 4.2.2 Part 2: Allocating the items to the agents

Next, we allocate the items in each bundle $B_j \in \mathbf{B}$ to the selected agents in $G_j$. Let $i$ be any agent in $G_j$ and $B'_j = B_j \setminus \{e_j^*\}$ where $e_j^*$ is the item lastly added to $B_j$. We first imaginatively assign the items in $B'_j$ to $i$'s bins as illustrated by Figure 1. We first put $i$'s large items in $B'_j$ into individual empty bins. Then we greedily put into the bins the remaining small items in $B'_j$ in decreasing order of their sizes, as long as the total size of the assigned items does not exceed the bin's capacity. The first time when the total size exceeds the capacity, we move to the next bin and so on (if all the bins with large items are filled, we move to an empty bin). We call the item lastly added to each bin that makes the total size exceed the capacity an *extra item*. Denote by $J_i(B'_j)$ the set of extra items and by $W_i(B'_j) = B'_j \setminus J_i(B'_j)$ the other items in $B'_j$.

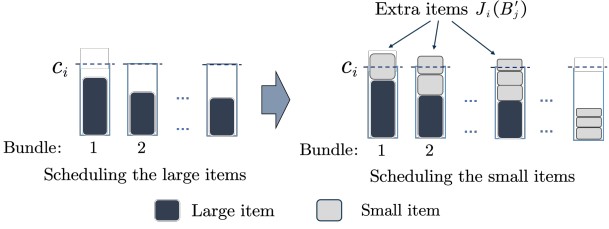

Figure 1: Imaginary assignment of $B'_j$ to agent $i$'s bins

If all items in $B_j$ are large for some agent $i \in G_j$, we allocate all of them to $i$. Otherwise, we know that round $j$ enters the bag filling procedure, thus there are two agents in $G_j$ and the last item $e_j^*$ is small for both of them. Letting $i_1$ be the agent who has more large items in $B_j$ and $i_2$ be the other agent, we allocate $i_1$ the items in $W_{i_1}(B'_j)$ and allocate $i_2$ the items in $J_{i_1}(B'_j) \cup \{e_j^*\}$.

Now we are ready to prove Theorem 3.

**Proof of Theorem 3.** Consider any round $j \in [d]$. If all items in $B_j$ are large for some agent $i \in G_j$, by Property 1, we know that there are at most $\mathsf{MMS}_i^d$ items in $B_j$. Thus $i$ can pack all items in $B_j$ using no more than $\mathsf{MMS}_i^d$ bins.

For the other case, recall that the agent $i_1 \in G_j$ who has more large items in $B_j$ receives the items in $W_{i_1}(B'_j)$, and the other agent $i_2$ receives the items in $J_{i_1}(B'_j) \cup \{e_j^*\}$. We first discuss agent $i_1$. By Property 1, we know that for each agent $i \in \{i_1, i_2\}$, there are at most $\mathsf{MMS}_i^d$ large items in $B'_j$ and $s_i(B'_j) \leq \frac{s_i(M)}{d}$. These two facts imply that in the imaginative assignment of $B'_j$ to $i_1$, no more than $\mathsf{MMS}_{i_1}^d$ bins are used. Since otherwise, $s_{i_1}(B'_j) > \mathsf{MMS}_{i_1}^d \cdot c_{i_1} \geq \frac{s_{i_1}(M)}{d}$, a contradiction. Therefore, $i_1$ can pack all items in $W_{i_1}(B'_j)$ using no more than $\mathsf{MMS}_{i_1}^d$ bins.

Next we discuss agent $i_2$. Observe that in the imaginary assignment of $B'_j$ to $i_1$, for each extra item in $J_{i_1}(B'_j)$, there exists another item in the same bin with a larger size. Therefore, we have

---

**Algorithm 3** Computing 2-MMS allocations for the bin packing setting

---

**Input:** An IDO bin packing instance $(N, M, \{v_i\}_{i \in N}, \{s_i\}_{i \in N})$.
**Output:** An allocation $\mathbf{A} = (A_1, \ldots, A_n)$ such that $v_i(A_i) \leq 2 \cdot \mathsf{MMS}_i$ for all $i \in N$.

 1: Initialize $R \leftarrow M$.
 2: **for** $j = 1$ to $n$ **do**
 3:     Initialize $B_j \leftarrow \emptyset$, $t \leftarrow j$, $k \leftarrow$ an arbitrary agent in $N$.
 4:     **while** $e_t \in R$ and there exists an agent $i \in N$ who thinks $e_t$ is large **do**
 5:         $B_j \leftarrow B_j \cup \{e_t\}$, $R \leftarrow R \setminus \{e_t\}$, $t \leftarrow t + n$.
 6:         $k \leftarrow i$.
 7:     **end while**
 8:     **while** there exists an agent $i \in N$ that satisfies $s_i(B_j) \leq \frac{s_i(M)}{n}$ and $L_i \cap R \neq \emptyset$ **do**
 9:         $k \leftarrow i$.
10:         Pick the smallest item $e \in R$, $B_j \leftarrow B_j \cup \{e\}$, $R \leftarrow R \setminus \{e\}$.
11:     **end while**
12:     $A_k \leftarrow B_j$, $N \leftarrow N \setminus \{k\}$.
13: **end for**

---

$s_{i_2}(J_{i_1}(B_j')) \leq \frac{s_{i_2}(B_j')}{2} \leq \frac{s_{i_2}(M)}{2d}$. Combining with the fact that there are at most $\mathsf{MMS}_{i_2}^d$ large items in $B_j'$ for $i_2$, we know that $i_2$ can use no more than $\mathsf{MMS}_{i_2}^d$ bins to pack all items in $J_{i_1}(B_j')$ and there exists one bin with at least half the capacity not occupied. Since otherwise, $s_{i_2}(J_{i_1}(B_j')) > \mathsf{MMS}_{i_2}^d \cdot \frac{c_{i_2}}{2} \geq \frac{s_{i_2}(M)}{2d}$, a contradiction. Recall that the last item $e_j^*$ is small for $i_2$, it can be put into the bin that has enough unoccupied capacity. Therefore, $i_2$ can also pack all items in $J_{i_1}(B_j') \cup \{e_j^*\}$ using at most $\mathsf{MMS}_{i_2}^d$ bins, which completes the proof. ∎

For the multiplicative relaxation of MMS, by Theorem 3 and Observation 1, a $\lceil \frac{n}{\lfloor \frac{n}{2} \rfloor} \rceil$-MMS allocation is guaranteed. Actually, we can slightly modify Algorithm 2 to compute a 2-MMS allocation.

**Corollary 2** *A 2-MMS allocation always exists for any bin packing instance.*

**Proof.** To compute a 2-MMS allocation, we replace the value of $d$ with $n$ in Algorithm 2 and select only one agent in each round who receives the bag in that round. The modified algorithm is formally presented in Algorithm 3. Following the same reasonings in Parts 1 and 2 (i.e., Subsubsections 4.2.1 and 4.2.2), it is not hard to see that all items can be allocated in Algorithm 3 and for any $i \in N$, there are at most $\mathsf{MMS}_i$ large items in $A_i$. Besides, if the last item $e_i^*$ is small for $i$, we have $s_i(A_i \setminus \{e_i^*\}) \leq \frac{s_i(M)}{n}$. Again, in the imaginary assignment of $A_i \setminus \{e_i^*\}$ to $i$, no more than $\mathsf{MMS}_i$ bins are used and at least one of them does not have an extra item. Therefore, agent $i$ can use $\mathsf{MMS}_i$ bins to pack all items in $W_i(A_i \setminus \{e_i^*\})$ and another $\mathsf{MMS}_i$ bins to pack all items in $J_i(A_i \setminus \{e_i^*\}) \cup \{e_i^*\}$, which completes the proof. ∎

In Appendix D.2, we show that the above multiplicative ratio is actually tight in the sense that there exists an instance where no allocation is better than 2-MMS. Besides, in Appendix D.3, we show that the algorithm that proves Corollary 2 actually computes an allocation where every agent $i$ can use at most $\frac{3}{2}\mathsf{MMS}_i + 1$ bins to pack all the items allocated to her.

## 5 Job scheduling setting

The second setting encodes a job scheduling problem where the items are jobs that need to be processed by the agents. Each item $e_j \in M$ has a size $s_{i,j} \geq 0$ to each agent $i \in N$, and for a set of items $S \subseteq M$, $s_i(S) = \sum_{e_j \in S} s_{i,j}$. As the bin packing setting, we only consider IDO instances where $s_{i,1} \geq \cdots \geq s_{i,m}$ for all $i$. Each agent $i \in N$ exclusively controls a set of $k_i$ machines $P_i = [k_i]$ with possibly different speed $\rho_{i,l}$ for each $l \in P_i$. Without loss of generality, we assume $\rho_{i,1} \geq \cdots \geq \rho_{i,k_i}$. Upon receiving a set of items $S \subseteq M$, agent $i$'s cost $v_i(S)$ is the minimum completion time of processing $S$ using her own machines $P_i$ (i.e., *the makespan of $P_i$*). Formally,

$$v_i(S) = \min_{(T_1, \ldots, T_{k_i}) \in \Pi_{k_i}(S)} \max_{l \in [k_i]} \frac{\sum_{e_t \in T_l} s_{i,t}}{\rho_{i,l}}.$$

Note that the computation of $v_i(S)$ is NP-hard if $k_i \geq 2$. For any two sets $S_1$ and $S_2$, $v_i(S_1 \cup S_2) \leq v_i(S_1) + v_i(S_2)$ since the makespan of scheduling $S_1 \cup S_2$ is no larger than the sum of the makespans of scheduling $S_1$ and $S_2$ separately, thus $v_i(\cdot)$ is subadditive.

Regarding the value of $\mathsf{MMS}_i^d$, intuitively, it is obtained by partitioning the items into $d \cdot k_i$ bundles, and allocating them to $k_i$ different types of machines (with possibly different speeds) where each type has $d$ identical machines so that the makespan is minimized.[3] Note that when each agent controls a single machine, i.e., $k_i = 1$ for all $i$, the problem degenerates to the additive cost case, and thus the job scheduling setting strictly generalizes the additive setting.

For the job scheduling setting, we have the following two main results.

**Theorem 4** *A 1-out-of-$\lfloor \frac{n}{2} \rfloor$ MMS allocation always exists for any job scheduling instance.*

**Corollary 3** *A 2-MMS allocation always exists for any job scheduling instance.*

Note that simply partitioning the items into $n$ bundles in a round-robin fashion does not guarantee 1-out-of-$\lfloor \frac{n}{2} \rfloor$ MMS even for the simpler additive cost setting. Consider an instance where there are four identical agents and five items with costs 4, 1, 1, 1, 1, respectively. For this instance, the value of 1-out-of-$\lfloor \frac{n}{2} \rfloor$ MMS for each agent is 4 as the 1-out-of-2 MMS defining partition is $\{\{4\}, \{1, 1, 1, 1\}\}$. However, the round-robin algorithm allocates two items with costs 4 and 1 to one agent, who receives a cost more than her 1-out-of-$d$ MMS. Our algorithm overcomes this problem by first partitioning the items into $\lfloor \frac{n}{2} \rfloor$ bundles and then carefully allocating the items in each bundle to two agents.

Let $d = \lfloor \frac{n}{2} \rfloor$. For each agent $i \in N$ and each machine $l \in P_i$, let $c_{i,l} = \rho_{i,l} \cdot \mathsf{MMS}_i^d$ denote $l$'s capacity. In a nutshell, our algorithm consists of three parts: in the first part, we partition all items into $d$ bundles in a round-robin fashion. In the second part, for each of the $d$ bundles and each agent, we present an imaginary assignment of the items in the bundle to the agent's machines. These imaginary assignments are used in the third part to guide the allocation of the items in each of the $d$ bundles to two agents, such that each agent can assign her allocated items to her machines in a way that the total workload on each machine does not exceed its capacity (in other words, each agent's cost is no more than her 1-out-of-$d$ MMS). We defer the detailed algorithms and proofs to Appendix E.

# 6 Conclusion

In this work, we study fair allocation of indivisible chores when the costs are beyond additive and the fairness is measured by MMS. There are many open problems and further directions. First, there are only existential results for $\min\{n, \lceil \log m \rceil\}$-MMS allocations in the general subadditive cost setting and 1-out-of-$\lfloor \frac{n}{2} \rfloor$ MMS allocations in the job scheduling setting. Polynomial-time algorithms that achieve the same results remain as open problems. Second, for the general subadditive cost setting and the bin packing setting, we provide the tight approximation ratios, but for the job scheduling setting, we only have a lower bound of $\frac{44}{43}$, which is inherited from the additive cost setting [20]. One immediate direction is to design better approximation algorithms or lower bound instances for the job scheduling setting. Third, in the appendix, we show that for the bin packing setting, there exists an allocation where every agent's cost is no more than $\frac{3}{2}$ times her MMS plus 1. We suspect that the multiplicative factor can be improved to 1. Fourth, for the job scheduling setting, we restrict us on the case of related machines in the current work, it is interesting to consider the general model of unrelated machines. As we have mentioned, the notion of collective maximin share fairness in the job scheduling setting can be viewed as a group-wise fairness notion, which could be of independent interest. Finally, we can investigate other combinatorial costs that can better characterize real-world problems.

# Acknowledgement

The authors are ordered alphabetically. This work is funded by NSFC under Grant No. 62102333, HKSAR RGC under Grant No. PolyU 25211321, and CCF-Huawei Populus Grove Fund.

---

[3]We provide another interpretation in Appendix E.1, which shows that the job scheduling setting uncovers new research directions for group-wise fairness.

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

# Appendix

## A  More related works

Besides proportionality, in another parallel line of research, envy-freeness and its relaxations, namely envy-free up to one item (EF1) and envy-free up to any item (EFX), are also widely studied. It was shown in [35] and [11] for goods and chores, respectively, that an EF1 allocation exists for the monotone combinatorial functions. However, the existence of EFX allocations is still unknown even with additive functions. Therefore, approximation algorithms were proposed in [2, 42] for additive functions and in [39, 16] for subadditive functions. We refer the readers to [3] for a detailed survey on fair allocation of indivisible items.

## B  Missing materials in preliminaries

### B.1  Impossibility result for general cost functions

We provide an example to show that no bounded approximation ratio can be achieved for general cost functions. Note that there exist simpler examples, but we choose the following one because it represents a particular combinatorial structure – minimum spanning tree. Let $G = (V, E)$ be a graph shown in the left sub-figure of Figure 2, where the vertices $V$ are the items that are to be allocated, i.e., $M = V$. There are two agents $N = \{1, 2\}$ who have different weights on the edges as shown in the middle and right sub-figures of Figure 2. The cost functions are measured by the minimum spanning tree in their received subgraphs. Particularly, for any $S \subseteq V$, $v_i(S)$ equals the weight of the minimum spanning tree on $G[S]$ – the induced subgraph of $S$ in $G$ – under agent $i$'s weights. Thus, $\mathsf{MMS}_i = 0$, for both $i = 1, 2$, where an MMS defining partition for agent 1 is $\{v_1, v_2\}$ and $\{v_3, v_4\}$ and that for agent 2 is $\{v_1, v_4\}$ and $\{v_2, v_3\}$. However, it can be verified that no matter how the vertices are allocated to the agents, there is one agent whose cost is at least 1, which implies that no bounded approximation is possible for general costs.

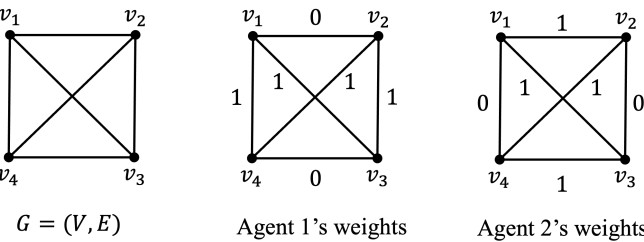

Figure 2: An instance with unbounded approximation ratio

### B.2  Proof of Observation 1

To prove the observation, it suffices to show $\mathsf{MMS}_i^d \leq \lceil \frac{n}{d} \rceil \cdot \mathsf{MMS}_i^n$ for any agent $i \in N$. Let $\mathbf{X} = (X_1, \ldots, X_n)$ be an MMS defining partition for agent $i$, which satisfies $v_i(X_j) \leq \mathsf{MMS}_i^n$ for every $j \in [n]$. Consider a $d$-partition $\mathbf{X}' = (X_1', \ldots, X_d')$ built by evenly distributing the $n$ bundles in $\mathbf{X}$ to the $d$ bundles in $\mathbf{X}'$; that is, the number of bundles distributed to the bundles in $\mathbf{X}'$ differs by at most one. Clearly, $\mathbf{X}'$ satisfies $v_i(X_j') \leq \lceil \frac{n}{d} \rceil \cdot \mathsf{MMS}_i^n$ for every $j \in [d]$. By the definition of 1-out-of-$d$ MMS, it follows that

$$\mathsf{MMS}_i^d \leq \max_{j \in [d]} v_i(X_j') \leq \lceil \frac{n}{d} \rceil \cdot \mathsf{MMS}_i^n,$$

thus completing the proof.

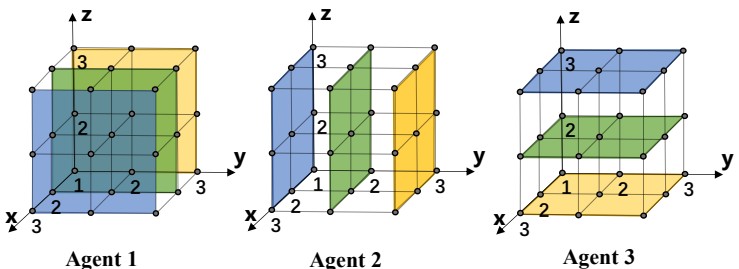

Figure 3: An instance with 3 agents and 27 items

## C  Missing materials in general subadditive cost setting

### C.1  An example that helps understand Theorem 1

The example is illustrated in Figure 3 where each agent has three covering planes. Take agent 1 for example, her three covering planes contain the items whose $x$ coordinates are 1, 2, 3, respectively. If there exists an allocation that is better than 3-MMS, then each agent is allocated items from at most 2 of her covering planes. Without loss of generality, we assume that agent 1 (or agents 2 and 3 respectively) is not allocated any item whose $x$ (or $y$ and $z$ respectively) coordinate is 1. Then, the item $(1, 1, 1)$ is not allocated to any agent, a contradiction.

### C.2  Proof of Corollary 1

We consider the same instance that is designed in Theorem 1. In this instance, we have proved that no matter how the items are allocated among the agents, there is at least one agent, say $i$, whose cost is $n$. Moreover, by the design of the cost functions, for any integer $d$, it can be observed that $\mathsf{MMS}_i^d = \lceil \frac{n}{d} \rceil$. Note that $\lceil \frac{n}{d} \rceil$ is always smaller than $n$ for all $d \geq 2$, thus the allocation is not 1-out-of-$d$ MMS to $i$.

## D  Missing materials in bin packing setting

### D.1  The IDO reduction

For a bin packing or job scheduling instance $I$, the IDO instance $I'$ is constructed by setting the size of each item $e_j \in M$ to each agent $i \in N$ in $I'$ to the $j$-th largest size of the items to $i$ in $I$. Then the IDO reduction is formally presented in the following lemma.

**Lemma 1** *For the bin packing or job scheduling setting, if there exists an allocation* $\mathbf{A}' = (A_1', \ldots, A_n')$ *in the IDO instance* $I'$ *such that* $v_i'(A_i') \leq \alpha \cdot \mathsf{MMS}_i^d(I')$ *for all* $i \in N$, *then there exists an allocation* $\mathbf{A} = (A_1, \ldots, A_n)$ *in the original instance* $I$ *such that* $v_i(A_i) \leq \alpha \cdot \mathsf{MMS}_i^d(I)$ *for all* $i \in N$.

**Proof.** We design Algorithm 4 that given $I$, $I'$ and $\mathbf{A}'$, computes the desired allocation $\mathbf{A}$. In the algorithm, we look at the items from $e_m$ to $e_1$. For each item, we let the agent who receives it in $I'$ pick her smallest unallocated item in $I$.

To prove the lemma, we first show that $v_i(A_i) \leq v_i'(A_i')$ for all $i \in N$. Consider the iteration where we look at the item $e_g$. We suppose that in this iteration agent $i$ picks item $e_{g'}$; that is, $e_g \in A_i'$, $e_{g'} \in A_i$ and $e_{g'}$ is the smallest unallocated item for $i$. Since an item is removed from the set $R$ after it is allocated, exactly $m - g$ items have been allocated before $e_{g'}$ is allocated. Therefore, $e_{g'}$ is among the top $m - g + 1$ smallest items for agent $i$. Recall that $e_g$ is the item with the exactly $(m - g + 1)$-th smallest size to $i$, hence $s_{i,g'} \leq s_{i,g}'$. The same reasoning can be applied to other items in $A_i'$ and $A_i$, and to other agents. It follows that for any $i \in N$, any $e_g \in A_i'$ and the corresponding $e_{g'} \in A_i$, $s_{i,g'} \leq s_{i,g}'$. For the bin packing or job scheduling setting, this implies $v_i(A_i) \leq v_i'(A_i')$. Since the maximin share depends on the sizes of the items but not on the order, the maximin share

of agent $i$ in $I'$ is the same as that in $I$, i.e., $\mathsf{MMS}_i^d(I') = \mathsf{MMS}_i^d(I)$. Hence, the condition that $v_i'(A_i') \leq \alpha \cdot \mathsf{MMS}_i^d(I')$ gives $v_i(A_i) \leq \alpha \cdot \mathsf{MMS}_i^d(I)$, which completes the proof. ∎

---

**Algorithm 4** IDO reduction for the bin packing and job scheduling settings

---

**Input:** A general instance $I$, the IDO instance $I'$ and an allocation $\mathbf{A}' = (A_1', ..., A_n')$ for the IDO instance such that $v_i'(A_i') \leq \alpha \cdot \mathsf{MMS}_i^d(I')$ for all $i \in N$.

**Output:** An allocation $\mathbf{A} = (A_1, ..., A_n)$ such that $v_i(A_i) \leq \alpha \cdot \mathsf{MMS}_i^d(I)$ for all $i \in N$.

1: For all $i \in N$ and $e_g \in A_i'$, set $p_g \leftarrow i$.
2: Initialize $A_i \leftarrow \emptyset$ for all $i \in N$, and $R \leftarrow M$.
3: **for** $g = m$ to $1$ **do**
4:     Pick $e_{g'} \in \arg\min_{e_k \in R}\{s_{p_g,k}\}$.
5:     $A_{p_g} \leftarrow A_{p_g} \cup \{e_{g'}\}, R \leftarrow R \setminus \{e_{g'}\}$.
6: **end for**

---

### D.2   Lower bound instance

We present an instance for the bin packing setting where no allocation can be better than 2-MMS. We first recall the impossibility instance given by Feige et al. [20]. In this instance there are three agents and nine items as arranged in a three by three matrix. The three agents' costs are shown in the matrices $V_1, V_2$ and $V_3$.

$$V_1 = \begin{pmatrix} 6 & 15 & 22 \\ 26 & 10 & 7 \\ 12 & 19 & 12 \end{pmatrix} \qquad\qquad V_2 = \begin{pmatrix} 6 & 15 & 23 \\ 26 & 10 & 8 \\ 11 & 18 & 12 \end{pmatrix}$$

$$V_3 = \begin{pmatrix} 6 & 16 & 22 \\ 27 & 10 & 7 \\ 11 & 18 & 12 \end{pmatrix}$$

Feige et al. [20] proved that for this instance the MMS value of every agent is 43, however, in any allocation, at least one of the three agents gets cost no smaller than 44.

We can adapt this instance to the bin packing setting and obtain a lower bound of 2. In particular, we also have three agents and nine items. The numbers in matrices $V_1, V_2$ and $V_3$ are the sizes of the items to agents 1, 2 and 3, respectively. Let the capacities of the bins be $c_i = 43$ for all $i \in \{1, 2, 3\}$. Accordingly, we have $\mathsf{MMS}_i = 1$ for all $i \in \{1, 2, 3\}$. Since in any allocation, there is at least one agent who gets items with total size no smaller than 44, for this agent, she has to use two bins to pack the assigned items, which means that no allocation can be better than 2-MMS.

### D.3   Computing $\frac{3}{2}\mathsf{MMS} + 1$ allocations

Recall that in the proof of Corollary 2, it has been shown that each agent $i \in N$ can use $\mathsf{MMS}_i$ bins to pack all items in $W_i(A_i \setminus \{e_i^*\})$ and another $\mathsf{MMS}_i$ bins to pack all items in $J_i(A_i \setminus \{e_i^*\}) \cup \{e_i^*\}$. Actually, since all items in $J_i(A_i \setminus \{e_i^*\}) \cup \{e_i^*\}$ are small for $i$ and at least two small items can be put into one bin, $i$ only needs $\lceil \frac{\mathsf{MMS}_i}{2} \rceil$ bins to pack all items in $J_i(A_i \setminus \{e_i^*\}) \cup \{e_i^*\}$. Therefore, each agent $i$ can use no more than $\frac{3}{2}\mathsf{MMS}_i + 1$ bins to pack all the items allocated to her.

## E   Missing materials in job scheduling setting

### E.1   Another interpretation to the job scheduling setting

An alternative way to explain the job scheduling setting is to view each agent $i$ as a group of $k_i$ small agents and $\mathsf{MMS}_i^d$ as the *collective maximin share* for these $k_i$ small agents. We believe this notion of collective maximin share is of independent interest as a group-wise fairness notion. We remark that this notion is different from the group-wise (and pair-wise) maximin share defined in [7] and [15], where the max-min value is defined for each single agent. In our definition, however, a set of agents share the same value for the items allocated to them.

### E.2 Algorithm

#### E.2.1 Part 1: partitioning the items into $d$ bundles

We first partition the items into $d$ bundles $\mathbf{B} = (B_1, \ldots, B_d)$ in a round-robin fashion. Specifically, we allocate the items in descending order of their sizes to the bundles by turns, from the first bundle to the last one. Each time, we allocate one item to one bundle, and when every bundle receives an item, we start over from the first bundle and so on. For any set of items $S$, let $S[l]$ be the $l$-th largest item in $S$, then the algorithm is formally presented in Algorithm 5.

---

**Algorithm 5** Partitioning the items into $d$ bundles

**Input:** An IDO job scheduling instance $(N, M, \{v_i\}_{i \in N}, \{s_i\}_{i \in N})$.
**Output:** A $d$-partition of $M$: $\mathbf{B} = (B_1, \ldots, B_d)$.

 1: Initialize $B_j \leftarrow \emptyset$ for every $j \in [d]$, and $r \leftarrow 1$.
 2: **while** $r \leq m$ **do**
 3:    **for** $j = 1$ to $d$ **do**
 4:      **if** $r \leq m$ **then**
 5:        $B_j \leftarrow B_j \cup \{M[r]\}$.
 6:        $r \leftarrow r + 1$.
 7:      **end if**
 8:    **end for**
 9: **end while**

---

By the characteristic of the round-robin fashion, we have the following important observation.

**Observation 2** *For each bundle $B_j \in \mathbf{B}$ and each item $e_k \in B_j \setminus \{B_j[1]\}$ (if exists), the $d-1$ items before $e_k$ (i.e., items $e_{k-1}, e_{k-2}, \ldots, e_{k-d+1}$) have at least the same sizes as $e_k$.*

#### E.2.2 Part 2: imaginary assignment

Next, for each bundle $B_j \in \mathbf{B}$ computed in the first part and each agent $i \in N$, we imaginatively assign the items in $B_j \setminus B_j[1]$ to $i$'s machines as follows. We greedily assign the items with larger sizes to $i$'s machines with faster speeds (in other words, with larger capacities), as long as the total workload on one machine does not exceed the its capacity. The first time when the workload exceeds the capacity, we move to the next machine and so on. The algorithm is formally presented in Algorithm 6 and illustrated in Figure 4. For each $l \in P_i$, $C_{i,l}^I$ contains the items imaginatively assigned to machine $l$ that do not make the total workload exceed $l$'s capacity, and $t_{i,l}$ is the last item assigned to $l$ that makes the total workload exceed the capacity. Note that $C_{i,l}^I$ may be empty and $t_{i,l}$ may be null. For simplicity, let $t_{i,0} = B_j[1]$; that is, $B_j[1]$ is assigned to an imaginary machine 0. The items in $\bigcup_{l \in [k_i]} C_{i,l}^I$ are called *internal items* (as shown by the *dark* boxes in Figure 4), and $\{t_{i,0}, \ldots, t_{i,k_i}\}$ are called *external items* (as shown by the *light* boxes).

---

**Algorithm 6** Imaginary assignment

**Input:** A bundle $B_j \in \mathbf{B}$ computed in the first part and an agent $i \in N$.
**Output:** Sets of internal items $\{C_{i,1}^I, \ldots, C_{i,k_i}^I\}$ and external items $\{t_{i,0}, \ldots, t_{i,k_i}\}$.

 1: Initialize $C_{i,l}^I \leftarrow \emptyset$, $t_{i,l} \leftarrow$ null for every $l \in [k_i]$, and $r \leftarrow 1$.
 2: **while** $r \leq |B_j|$ **do**
 3:    **for** $l = 1$ to $k_i$ **do**
 4:      $t_{i,l-1} \leftarrow B_j[r]$, $r \leftarrow r + 1$.
 5:      **while** $r \leq |B_j|$ and $s_i(C_{i,l}^I \cup \{B_j[r]\}) \leq c_{i,l}$ **do**
 6:        $C_{i,l}^I \leftarrow C_{i,l}^I \cup \{B_j[r]\}$, $r \leftarrow r + 1$.
 7:      **end while**
 8:    **end for**
 9: **end while**

---

For each bundle $B_j \in \mathbf{B}$ and each agent $i \in N$, the imaginary assignment has the following important properties.

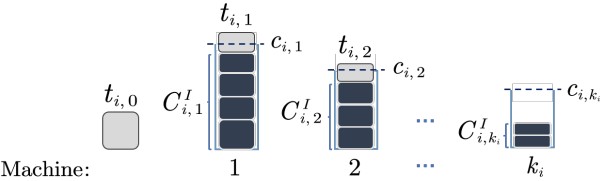

Figure 4: The imaginary assignment of $B_j$ to agent $i$

- **Property 1**: all items in $B_j \setminus \{B_j[1]\}$ can be assigned to agent $i$'s machines. Besides, the last machine $k_i$ does not have an external item; that is, $t_{i,k_i}$ is null.

- **Property 2**: for any $1 \leq l \leq k_i$, the total size of the internal items $C_{i,l}^I$ does not exceed the capacity of machine $l$, i.e., $s_i(C_{i,l}^I) \leq c_{i,l}$

- **Property 3**: for any $1 \leq l \leq k_i$, the external item $t_{i,l-1}$ (if not null) has size no larger than the capacity of machine $l$, i.e., $s_i(t_{i,l-1}) \leq c_{i,l}$.

**Proof.** The first property holds since otherwise, $s_i(B_j \setminus \{B_j[1]\}) > \sum_{l \in [k_i]} c_{i,l}$. By Observation 2, it follows that

$$s_i(M) > d \cdot s_i(B_j \setminus \{B_j[1]\}) > d \cdot \sum_{l \in [k_i]} c_{i,l}.$$

However, since all items can be assigned to $i$'s machines in $i$'s 1-out-of-$d$ MMS defining partition, we have $s_i(M) \leq d \cdot \sum_{l \in [k_i]} c_{i,l}$, a contradiction.

The second property directly follows the algorithm. For the third property, $s_i(t_{i,0}) \leq c_{i,1}$ follows two facts that $t_{i,0}$ is assigned to some machine in $i$'s 1-out-of-$d$ MMS defining partition and $c_{i,1}$ is the largest capacity of the machines. We then consider $l \in [k_i - 1]$ and show $s_i(t_{i,l}) \leq c_{i,l+1}$ (if $t_{i,l}$ is not null). The same reasoning can be applied to any other $l' \in [k_i - 1]$. Let $S_1 = \bigcup_{p \in [l]} (C_{i,p}^I \cup \{t_{i,p}\})$. From the algorithm, we know that $s_i(S_1) > \sum_{p \in [l]} c_{i,p}$ and $t_{i,l}$ is the smallest item in $S_1$. By Observation 2, there exist another $d - 1$ disjoint sets of items $\{S_2, \ldots, S_d\}$ such that $s_i(S_k) \geq s_i(S_1)$ for every $k \in [2, d]$ and $t_{i,l}$ is also the smallest item in $\bigcup_{k \in [d]} S_k$. Hence, $\sum_{k \in [d]} s_i(S_k) > d \cdot \sum_{p \in [l]} c_{i,p}$. This implies that in $i$'s 1-out-of-$d$ MMS defining partition, at least one item in $\bigcup_{k \in [d]} S_k$ is assigned to machine $p \geq l + 1$. Combining with the fact that $t_{i,l}$ is the smallest item in $\bigcup_{k \in [d]} S_k$, we have $s_i(t_{i,l}) \leq c_{i,l+1}$. ∎

By these properties, for each machine $l \in P_i$, we can assign either the internal items $C_{i,l}^I$ or the external item $t_{i,l-1}$ to $l$, such that its completion time does not exceed $\mathsf{MMS}_i^d$. This intuition guides the allocation of the items to the agents in the following part.

### E.2.3  Part 3: allocating the items to the agents

Lastly, for any bundle $B_j \in B$, we arbitrarily choose two agents $i_1, i_2 \in N$ and allocate them the items in $B_j$ as formally described in Algorithm 7. Recall that in the imaginary assignment of $B_j$ to each agent $i \in \{i_1, i_2\}$, the items in $B_j$ are divided into internal items $\bigcup_{l \in [k_i]} C_{i,l}^I$ and external items $\{t_{i,0}, \ldots, t_{i,k_i}\}$. Let $E = \{e_1^*, \ldots, e_{|E|}^*\}$ contain all external items shared by $i_1$ and $i_2$. Note that $e_1^* = t_{i_1,0} = t_{i_2,0}$. We allocate the items in $B_j$ to agents $i_1$ and $i_2$ in $|E|$ rounds. In each round $q \in [|E|]$, we first find the machines of $i_1$ and $i_2$ to which the shared external items $e_q^*$ and $e_{q+1}^*$ are assigned (denoted by $l_1, l_2, l_1'$ and $l_2'$, respectively. If $q = |E|$, simply let $l_1' = k_{i_1}$ and $l_2' = k_{i_2}$). We then find the agent $i_k \in \{i_1, i_2\}$ whose machine $l_k + 1$ has more internal items. We allocate $i_k$ her internal items from machine $l_k + 1$ to machine $l_k'$, and allocate the other agent $i_{\bar{k}}$'s external items from machine $l_{\bar{k}}$ to machine $l_{\bar{k}}' - 1$.

Since $2 \cdot d = 2 \cdot \lfloor \frac{n}{2} \rfloor \leq n$, no more than $n$ agents are needed to allocate all items. Thus to prove Theorem 4, it remains to show that each agent can assign her allocated items to her machines such that the total workload on each of the machines does not exceed its capacity.

---
**Algorithm 7** Allocating the items to the agents
---
**Input:** A $d$-partition of the items $\mathbf{B} = (B_1, \ldots, B_d)$ returned by Algorithm 5.

**Output:** An allocation $\mathbf{A} = (A_1, \ldots, A_n)$ such that $v_i(A_i) \leq \mathsf{MMS}_i^d$ for all $i \in N$.

1: Initialize $A_i \leftarrow \emptyset$ for every $i \in N$.
2: **for** $j = 1$ to $d$ **do**
3:      Arbitrarily choose 2 agents $i_1, i_2 \in N$, $N \leftarrow N \setminus \{i_1, i_2\}$.
4:      $\{C_{i_1,1}^I, \ldots, C_{i_1,k_{i_1}}^I\}, \{t_{i_1,0}, \ldots, t_{i_1,k_{i_1}}\} \leftarrow$ *Algorithm* 6$(B_j, i_1)$.
5:      $\{C_{i_2,1}^I, \ldots, C_{i_2,k_{i_2}}^I\}, \{t_{i_2,0}, \ldots, t_{i_2,k_{i_2}}\} \leftarrow$ *Algorithm* 6$(B_j, i_2)$.
6:      $E \leftarrow \{t_{i_1,0}, \ldots, t_{i_1,k_{i_1}}\} \cap \{t_{i_2,0}, \ldots, t_{i_2,k_{i_2}}\}$. Re-label $E \leftarrow \{e_1^*, \ldots, e_{|E|}^*\}$. // Shared external items by $i_1$ and $i_2$
7:      **for** $q = 1$ to $|E|$ **do**
8:         Find $l_1 \in [0, k_{i_1}]$ and $l_2 \in [0, k_{i_2}]$ such that $e_q^* = t_{i_1,l_1} = t_{i_2,l_2}$.
9:         **if** $q < |E|$ **then**
10:            Find $l_1' \in [0, k_{i_1}]$ and $l_2' \in [0, k_{i_2}]$ such that $e_{q+1}^* = t_{i_1,l_1'} = t_{i_2,l_2'}$.
11:         **else**
12:            $l_1' = k_{i_1}$ and $l_2' = k_{i_2}$.
13:         **end if**
14:         **if** $|C_{i_1,l_1+1}^I| \geq |C_{i_2,l_2+1}^I|$ **then**
15:            $A_{i_1} \leftarrow \bigcup_{l=l_1+1}^{l_1'} C_{i_1,l}^I$, $A_{i_2} \leftarrow \bigcup_{l=l_1}^{l_1'-1} t_{i_1,l}$.
16:         **else**
17:            $A_{i_2} \leftarrow \bigcup_{l=l_2+1}^{l_2'} C_{i_2,l}^I$, $A_{i_1} \leftarrow \bigcup_{l=l_2}^{l_2'-1} t_{i_2,l}$.
18:         **end if**
19:      **end for**
20: **end for**
---

**Proof of Theorem 4.** Consider any bundle $B_j \in \mathbf{B}$ and assume the two chosen agents are $i_1, i_2 \in N$. We first look at the first round of the process of allocating the items in $B_j$ to $i_1$ and $i_2$. Without loss of generality, assume that the first machine of $i_1$ contains more internal items than that of $i_2$, i.e., $C_{i_1,1}^I \geq C_{i_2,1}^I$. From the algorithm, the items $i_1$ takes are $\bigcup_{l=1}^{l_1'} C_{i_1,l}^I$. By the second property of the imaginary assignment, these items can be assigned to the first $l_1'$ machines of $i_1$ such that the total workload on each machine does not exceed its capacity. Besides, the items $i_2$ takes are $\bigcup_{l=0}^{l_1'-1} t_{i_1,l}$, which are $e_1^*$ and a subset of $\bigcup_{l=2}^{l_2'} C_{i_2,l}^I$. By the second and third properties of the imaginary assignment, these items can be assigned to the first $l_2'$ machines of $i_2$ such that the total workload on each machine does not exceed its capacity. The same reasoning can be applied to all following rounds. By induction, it follows that both $i_1$ and $i_2$ can assign their allocated items to their machines such that the total workload on each machine does not exceed its capacity. This means that both $i_1$ and $i_2$ receive costs no more than their 1-out-of-$d$ MMS, which completes the proof. ∎

For the multiplicative relaxation of MMS, by Theorem 4 and Observation 1, a $\lceil \frac{n}{\lfloor \frac{n}{2} \rfloor} \rceil$-MMS allocation is guaranteed. As the bin packing setting, after a slight modification, Algorithm 5 computes a 2-MMS allocation, which is better than $\lceil \frac{n}{\lfloor \frac{n}{2} \rfloor} \rceil$-MMS.

**Proof of Corollary 3.** We show that by replacing the value of $d$ with $n$, Algorithm 5 computes a 2-MMS allocation. Particularly, in the new version of Algorithm 5, we partition the items in $M$ into $n$ bundles in a round-robin fashion and allocate each of the $n$ bundles to one agent in $N$. By the properties of the imaginary assignment, for each agent, the makespan of processing either the internal items or the external items in her bundle using her machines does not exceed $\mathsf{MMS}_i^n$. This implies that for each agent, the cost of her bundle does not exceed $2 \cdot \mathsf{MMS}_i^n$, which completes the proof. ∎

# F   Proportionality up to one or any item

We now discuss two other relaxations for proportionality, i.e., proportional up to one item (PROP1) and proportional up to any item (PROPX), which are also widely studied for additive costs.

**Definition 2 ($\alpha$-PROP1 and $\alpha$-PROPX)** *An allocation* $\mathbf{A} = (A_1, \ldots, A_n)$ *is $\alpha$-approximate proportional up to one item ($\alpha$-PROP1) if* $v_i(A_i \backslash \{e\}) \leq \alpha \cdot \frac{v_i(M)}{n}$ *for all agents* $i \in N$ *and some item* $e \in A_i$. *It is $\alpha$-approximate proportional up to any item ($\alpha$-PROPX) if* $v_i(A_i \backslash \{e\}) \leq \alpha \cdot \frac{v_i(M)}{n}$ *for all agents* $i \in N$ *and any item* $e \in A_i$. *The allocation is PROP1 or PROPX if* $\alpha = 1$.

It is easy to see that a PROPX allocation is also PROP1. Although exact PROPX or PROP1 allocations are guaranteed to exist for additive costs, when the costs are subadditive, no algorithm can be better than $n$-PROP1 or $n$-PROPX. Consider an instance with $n$ agents and $n + 1$ items. The cost function is $v_i(S) = 1$ for all agents $i \in N$ and any non-empty subset $S \subseteq M$. Clearly, the cost function is subadditive since $v_i(S) + v_i(T) \geq v_i(S \cup T)$ for any $S, T \subseteq M$. By the pigeonhole principle, at least one agent $i$ receives two or more items in any allocation of $M$. After removing any item $e \in A_i$, $A_i$ is still not empty. That is, $v_i(A_i \backslash \{e\}) = 1 = n \cdot \frac{v_i(M)}{n}$ for any $e \in A_i$. This example can be easily extended to the bin packing and job scheduling settings, and thus we have the following theorem.

**Theorem 5** *For the bin packing and job scheduling settings, no algorithm performs better than $n$-PROP1 or $n$-PROPX.*

**Proof.** For the bin packing setting, consider an instance with $n$ agents and $n + 1$ items. The capacity of each agent's bins is 1, i.e, $c_i = 1$ for all $i \in N$. Each item is very tiny so that every agent can pack all items in just one bin, e.g., $s_{i,j} = \frac{1}{n+1}$ for any $i \in N$ and $e_j \in M$. Therefore, we have $v_i(M) = 1$ and $\mathsf{PROP}_i = \frac{1}{n}$ for each agent $i \in N$. By the pigeonhole principle, at least one agent $i$ receives two or more items in any allocation of $M$. After removing any item $e \in A_i$, agent $i$ still needs one bin to pack the remaining items. Hence, we have $v_i(A_i \backslash \{e\}) = 1 = n \cdot \mathsf{PROP}_i$ for any $e \in A_i$.

For the job scheduling setting, consider an instance with $2n$ agents and $2n + 1$ items where each agent possesses $2n$ machines with the same speed of 1, and the size of each item is 1 for every agent. It can be easily seen that for every agent $i \in N$, the maximum completion time of her machines is minimized when assigning two items to one machine and one item to each of the remaining $2n - 1$ machines. Therefore, we have $v_i(M) = 2$ and $\mathsf{PROP}_i = \frac{2}{2n} = \frac{1}{n}$ for any $i \in N$. Similarly, by the pigeonhole principle, at least one agent $i$ receives two or more items in any allocation of $M$. This implies that $v_i(A_i \backslash \{e\}) = 1 = n \cdot \mathsf{PROP}_i$ for any $e \in A_i$, thus completing the proof. $\blacksquare$

