# OpenReview forum: "Fair Allocation of Indivisible Chores: Beyond Additive Costs"
_NeurIPS.cc/2023/Conference — NeurIPS 2023 poster_

### Official Review · Reviewer_6vH3 · 2023-06-22

**Soundness:** 4 excellent
**Presentation:** 3 good
**Contribution:** 2 fair
**Rating:** 5
**Confidence:** 4

**Summary:**

This paper studies the allocation of m indivisible chores among n agents with non-additive preferences. The authors show that, for the case of approximate MMS, the best approximation factor is super constant, and specifically they give a lower bound of min{n, log m/ log log m } for submodular costs, and an upper bound of min{n, log m} for subadditive costs. The lower bound also implies a negative result for 1-out-of-d MMS allocations in this setting.
The authors proceed to study special cases of subadditive costs, and specifically costs encoded by combinatorial problems, and namely bin-packing and job scheduling. So, for example, in the case of bin-packing, the cost for a subset of items is the minimum number of bins to pack them. For both cases, the authors give an algorithm for finding a 1-out-of-(n/2) MMS allocation.

**Strengths:**

- The paper studies an interesting, natural problem: chore allocation beyond additive costs.
- The results are relatively complete, and the authors settle, up-to-constants, all their questions.


**Weaknesses:**

- The main algorithmic results (for bin-packing and scheduling) seem like twists to the standard approximation algorithms (e.g., Next Fit or Best Fit for bin-packing). Of course, this is expected, but these connections/insights are not explicit in the text, so it’s harder to see what’s new about this work.

(Some typos:
Line 134: “which is somehow the most unfair algorithm” -> drop “somehow”.
Line 342: “or the job scheduling setting, we restrict us on the case”)

**Questions:**

Have combinatorial values of this nature (and as a tool for bypassing negative results) been considered in the goods case?

---

> ### Author Rebuttal · Authors · 2023-08-09
>
> We thank the reviewer for your appreciation of the problem we have studied and the results we have obtained. We also thank the reviewer for the constructive and helpful comments.
>
> **Question 1: combinatorial problems considered in the case of goods**
>
> **Response:** We appreciate this question and will incorporate the following discussion in the revised paper, as we consider it a compelling motivation for our research problems.
>
> When it comes to goods, there have been several papers that delve into concrete combinatorial problems and seek to improve approximation ratios compared to more general valuations. For instance, we draw attention to two papers [1,2] that introduce interval scheduling and independent set structures, respectively, into MMS fair allocation problems. In both cases, the induced valuation functions correspond to special cases of XoS (fractionally subadditive) functions. While a general XoS valuation guarantees an approximation ratio of 1/4.6, these two works [1,2] manage to enhance this approximation for their specific functions.
>
> [1] “Fair Scheduling for Time-dependent Resources” at NeurIPS 2021
>
> [2] “Fair Allocation of Conflicting Items” at AAMAS 2022
>
> [3] “Improved Maximin Guarantees for Subadditive and Fractionally Subadditive Fair Allocation Problem” at AAAI 2022
>
> Some other combinatorial problems that have been studied for goods include the knapsack problem and matroid constraints, e.g.,
>
> [4] “Approximation Algorithm for Computing Budget-Feasible EF1 Allocations” at AAMAS 2023
>
> [5] “Guaranteeing Envy-Freeness under Generalized Assignment Constraints” at EC 2023
>
> [6] “On fair division under heterogeneous matroid constraints” at JAIR 2023
>
>
> **Comment 1: connections with standard algorithms**
>
> **Response:** Yes, our algorithms reflect some ideas from standard algorithms (such as Next Fit), and we will make the connections and differences more explicit. Informally, in the second phase of our algorithms, we prove that we can assign each bundle obtained from the first phase to two agents without violating the MMS constraints. To prove this claim, we introduce an imaginary partition of the items, which is derived from the standard algorithms with certain modifications, such as allowing for one over-filled item. By leveraging the structural properties of this partition, we can prove our claim. In essence, the standard algorithms serve as a step within our algorithms.

---

> > ### Comment · Reviewer_6vH3 · 2023-08-15
> >
> > Thank you for your reply. I do not have any other questions at this stage.

---

### Official Review · Reviewer_AniE · 2023-07-05

**Soundness:** 3 good
**Presentation:** 3 good
**Contribution:** 3 good
**Rating:** 6
**Confidence:** 4

**Summary:**

This paper studies the MMS fair allocation of combinatorial tasks (indivisible chores) problem where the cost function is submodular or subadditive. For submodular functions, they prove that no algorithm can ensure better than min{n, log m/log log m}-approximation. For more general subadditive cost functions, they prove that there always exists an allocation that is min{n , log m}-approximation MMS, which is (almost) asymptotically tight. What’s more, for ordinal relaxation, 1-out-of-d MMS, they prove that for any d≥2, there is an instance for which no allocation is 1-out-of-d MMS. Finally, the authors give two specific settings which are bin packing setting and job scheduling setting and prove that 1-out-of-[n/2] MMS allocations always exist for these two settings.

**Strengths:**

This paper provides solid and clean theoretical results on MMS chores allocations. It also contains some interesting techniques. For example, the author gives a quite interesting example for Theorem 1, and I also find the proof of Theorem 2 mathematically natural and complete.

**Weaknesses:**

I do not find any obvious weaknesses. Perhaps the relevance of this paper to machine learning is not very strong.

**Questions:**

No question.

**Limitations:**

Not applicable.

---

> ### Author Rebuttal · Authors · 2023-08-09
>
> We express our gratitude to the reviewer for the dedicated effort in reviewing our paper. We are especially appreciative of the reviewer's acknowledgment of the theoretical results and techniques presented in our paper. We also understand the reviewer’s concern about the relevance of our paper to machine learning. We humbly offer the following justifications, hoping they will be helpful in addressing this concern.
>
> On one hand, we can see that fair division works have been increasingly welcomed at conferences such as NeurIPS and ICML in recent years. For example, a growing number of fair allocation papers have been presented at NeurIPS and ICML in recent years, e.g.,
>
> -  “Fair Scheduling for Time-dependent Resources” at NeurIPS 2021
> -  “Fair and Efficient Allocations Without Obvious Manipulations” at NeurIPS 2022
> -  “Multi-agent Online Scheduling: MMS Allocations for Indivisible Items” at ICML 2023
>
> On the other hand, our paper focuses on submodular (and subadditive) functions, and these functions have significant relevance to various machine learning and optimization scenarios, as we attempted to motivate the relevance in the introduction. Thus, we think the investigation of fair division under submodular functions can contribute to NeurIPS.

---

> > ### Comment · Reviewer_AniE · 2023-08-18
> >
> > Thanks a lot for your response. I do not have any further questions.

---

### Official Review · Reviewer_RP1j · 2023-07-05

**Soundness:** 3 good
**Presentation:** 3 good
**Contribution:** 3 good
**Rating:** 7
**Confidence:** 4

**Summary:**

The paper deals with problem of allocating indivisible chores to agents whose valuation functions for bundles of chores are subadditive or submodular, where an allocation that guarantees every agent their maximin share (MMS) may not exist. Here, an agent's maximin share is the agent's worst-case disutility from a partitioning of the items that minimizes the disutility of the worst (maximum disutility) partition. The paper provides new upper and lower bounds on the approximability of MMS allocations.

I have reviewed a previous version of this paper submitted to IJCAI 23 where I was positive about the technical contributions of the paper. This revision addresses the concerns with writing and a minor technical issue raised there. The revisions have certainly helped with the readability.

**Strengths:**

The problem setting of subadditive and submodular (dis)utility functions is novel.

The new results for these classes of valuation functions provided is this paper are likely to be of interest to the comsoc / fair division research community.

The technical results are interesting, non-trivial and use interesting techniques that are new to me. I was able to verify the technical results.

Relevant related work is well cited and discussed.

**Weaknesses:**

The relevance to NeurIPS for what seems a very AI/computational social choice focused paper is not clear although this is attempted in the introduction.



**Questions:**

None

---

> ### Author Rebuttal · Authors · 2023-08-09
>
> We thank the reviewer for the time and effort in reviewing our paper again. We are pleased to hear that the reviewer acknowledges the technical contribution of our work and the improvements made in the revised manuscript. We will incorporate all reviewers’ suggestions to further enhance and refine the paper.
>
> We understand the reviewer’s concern about the relevance of our paper to NeurIPS. We humbly offer the following justifications, hoping they will be helpful in addressing this concern. On one hand, we can see that AI/Computational Social Choice works have been increasingly welcomed at conferences such as NeurIPS and ICML in recent years. For example, a growing number of fair allocation papers have been presented at NeurIPS and ICML in recent years, e.g.,
>
> - “Fair Scheduling for Time-dependent Resources” at NeurIPS 2021
> - “Fair and Efficient Allocations Without Obvious Manipulations” at NeurIPS 2022
> -  “Multi-agent Online Scheduling: MMS Allocations for Indivisible Items” at ICML 2023
>
> On the other hand, our paper focuses on submodular (and subadditive) functions, and these functions have significant relevance to various machine learning and optimization scenarios, as we attempted to motivate the relevance in the introduction. Thus, we think the investigation of fair division under submodular functions can contribute to NeurIPS.

---

> > ### Comment · Reviewer_RP1j · 2023-08-15
> >
> > Thank you for the response. I do not have additional questions at this time.

---

### Official Review · Reviewer_MebF · 2023-07-05

**Soundness:** 3 good
**Presentation:** 3 good
**Contribution:** 3 good
**Rating:** 5
**Confidence:** 3

**Summary:**

The paper studies the classical fair allocation of indivisible items setting with two twists: (1) Items correspond to *tasks* instead of *goods*, i.e., any agent would prefer receiving no item at all. (2) Valuations are not additive but may be subadditive. The paper also considers some special cases of subadditive valuations, namely, submodular, "bin packing", and "scheduling" costs. As an objective, the paper focusses on the classic notion of the maximin share (MMS). Since MMS allocations do not always exist, the paper considers both a multiplicative as well as an "ordinal" relaxation of the MMS notion.

The contribution of the paper is threefold (I am omitting the ordinal approximation results for simplicity): (1) For the subadditive case, the paper presents a lower bound of $\min\{n,log(m)/loglog(m)\}$ as well as a mechanism providing an approximation of $\min\{n,\log(m)\}$. (2) For bin packing, the paper presents a multiplicative $2$-approximation and a tight lower bound of $2$ for any mechanism. (3) For job scheduling, the paper presents a mechanism providing a multiplicative $2$-approximation, however, without a matching lower bound.

**Strengths:**

- The paper makes significant progress within the classic setting of fair allocation of indivisible items. While this literature has been long focussed on the case of additive valuations/costs, in recent years there has been a growing body of literature studying more general valuations/costs, with the paper under review being one of these. Hence, I am optimistic that the paper will lead to follow-up work.
- The paper develops new mechanisms that are tailored to the studied cost functions. These mechanisms and their analysis is certainly non-trivial and lead to a significant technical contribution.


**Weaknesses:**

- Since the result for the general, subadditive case is rather negative (i.e., there is no constant approximation for MMS), the main contribution of the paper is for the special cases of bin packing and scheduling. Having said this, this can hardly be seen as a critique for the paper, but rather as a sign of the challenging endeavor to study beyond additive costs.
- I think that the writing of the paper could be improved, as I had to reread several parts of the paper. I added a list of suggestions within the minor comments.
- Unfortunately, the newly developed mechanism is not very elegant, and one can't help but wonder whether there exists a simpler mechanism to achieve the same approximation guarantee. Also, the mechanisms do not come with a polynomial-time implementation, hence, leaving the question open whether the same guarantees can be achieved efficiently.

**Minor Comments**
- The paper uses the term "tasks", while a large fraction of the literature uses the term "chores". I would suggest to add a comment on that.
- line 21: You mention "functions" without clarifying their role in the problem. (Of course this is clear for any person knowing fair allocation, but for others it may not.)
- line 60: "As far as we know, all the above works also assume additive costs" - Sounds a bit weird in this context, since checking these papers should be doable.
- line 71: "the asymptotically tight multiplicative" - I think it is weird to use this phrase in a theorem environment, especially since you are ignoring log-factors. I would suggest to just mention the upper and lower bounds.
- line 99: "Note that no bounded approximation" - At this point, it is not clear what should be approximated.
- Proof of theorem 1: I was very confused of the usage of the term "covering planes" since, as far as I understand, these objects are actually (partial) grids, i.e. finite set of points.
- line 214: I think it would be helpful for the reader to learn about the meaning of the abbreviation "IDO".
- line 228-236: I did not find the intuition for the algorithm very helpful before reading the algorithm (and even after that). I would suggest to refine this, having in mind that the reader has not read the algorithm at this point.
- line 316: I think that $j \in P_i$ is a bad choice for an index since here $j$ is a machine but before $j$ used to correspond to jobs/items.
- Section 5: It would have been nice to hear some (very brief) summary of how Theorem 4 is achieved, i.e., how does the mechanism look like?

**Questions:**

-  Intuitively, subadditive costs make the problem "easier" in the sense that allocation all tasks to one agent at least has the same approximation guarantee as in the additive case. This is certainly not the case for superadditive cost functions. Do you have any results in this direction?
- I was missing a concrete (real-world) motivation for subadditive cost functions in the context of allocation tasks. Could you elaborate on that?

**Limitations:**

The limitations of the paper are well addressed, in particular, within Section 6. Here, the paper transparently communicates all resulting open questions.

---

> ### Author Rebuttal · Authors · 2023-08-09
>
> We thank the reviewer for appreciating our paper’s contribution to the literature on fair allocation, as well as the technical depth and the potential to stipulate subsequent research. We also thank the reviewer for pointing out our paper’s weaknesses and offering many constructive and helpful suggestions for improving our paper.
>
> In the following, we first answer your questions.
>
> **Question 1: on the superadditive cost functions.**
>
> **Response:** Yes, we have considered superadditive cost functions and found that no bounded approximation is possible. Let us consider a simple instance with two agents $N = \\{1, 2\\}$ and four items $M = \\{a, b, c, d\\}$. The cost functions are described below:
> - When $|S| \le 1$, $v_i(S) = 0$ for both $i = 1$ and $2$
> - When $|S| = 2$,
>    - $v_1(\\{a, d\\}) = v_1(\\{b, c\\}) = 0$; $v_1(\\{a, b\\}) = v_1(\\{a, c\\}) = v_1(\\{b, d\\}) = v_1(\\{c, d\\}) = 1$.
>    - $v_2(\\{a, b\\}) = v_2(\\{c, d\\}) = 0$; $v_2(\\{a, c\\}) = v_2(\\{a, d\\}) = v_2(\\{b, c\\}) = v_2(\\{b, d\\}) = 1$.
> - When $|S| = 3$, $v_i(S) = 1$ for both $i = 1$ and $2$
> - When $|S| = 4$, $v_i(S) = 2$ for both $i = 1$ and $2$
>
> It can be verified that the cost functions are superadditive. Besides, MMS$_i = 0$ for both $i = 1$ and $2$ since agent $1$ can partition the items into $\\{\\{a, d\\}, \\{b, c\\}\\}$ and agent $2$ can partition the items into $\\{\\{a, b\\}, \\{c, d\\}\\}$. However, no matter how the items are allocated to the agents, there is one agent whose cost is at least $1$.
>
> We have included a similar example in Appendix B.1 of the manuscript.
>
> **Question 2: concrete motivation for subadditive cost functions in the context of allocating tasks**
>
> **Response:** Firstly, we can regard the job scheduling and bin packing models as two examples that motivate subadditive cost functions in the context of allocating tasks. One interpretation of the cost of a set of tasks is the time taken to complete them. Sometimes agents may be able to perform tasks in parallel, in which scenario the agents have job-scheduling cost functions. Sometimes agents have to finish tasks by a deadline and can hire processors to finish them. The cost of hiring processors equals the number of processors, in which scenario the agents have job-scheduling cost functions.
>
> Secondly, and more concretely, doing laundry and cleaning the house can be carried out simultaneously, and thus induce subadditive cost functions. Similarly, when it comes to teaching, adding more students to a class can actually result in a decrease in the marginal cost for the teacher, as the teacher only needs to prepare the teaching materials once, regardless of the number of students.
>
> Next, we respond to your comments.
>
> **Comment 1: on the inherent difficulty of the problems.**
>
> **Response:** We thank the reviewer for understanding the inherent difficulty of the problems.
>
> **Comment 2: the writing of the paper could be improved**
>
> **Response:** We thank the reviewer for pointing out this weakness and offering so many helpful suggestions for improving the writing quality. We will carefully polish our paper and address the reviewer’s concerns.
>
> **Comment 3: simpler mechanisms to achieve the same approximation guarantee**
>
> **Response:** We have tried several simpler algorithms but none of them performs well, especially for the ordinal approximation settings. For example, round-robin is a commonly-used algorithm in fair division that guarantees $2$-MMS (multiplicative approximation) for additive cost functions. However, it does not ensure $1$-out-of-$\lfloor n/2 \rfloor$ MMS in terms of ordinal approximations. To see this, we can consider an instance with four identical agents and five items of costs $4,1,1,1,1$. A $1$-out-of-$\lfloor n/2 \rfloor$ MMS defining partition is $\\{\\{4\\},\\{1,1,1,1\\}\\}$ and thus the $1$-out-of-$\lfloor n/2 \rfloor$ MMS equals $4$. The round-robin algorithm allocates two items with costs $4$ and $1$ to one agent, who receives a cost higher than her $1$-out-of-$2$ MMS.
>
> This example also appears in Appendix E.2 of the manuscript.

---

> > ### Comment · Reviewer_MebF · 2023-08-14
> > **Response**
> >
> > I thank the authors for their detailed response. I do not have any further questions at this point.

---

### Decision · Program_Chairs · 2023-09-21

**Decision:**

Accept (poster)

**Comment:**

All reviewers felt that it is a solid fair division paper with solid theoretical results, though no reviewer seems to be extremely excited about them (and one reviewer commented that they are mostly based on existing techniques). Multiple reviewers raised the question about the relevance to NeurIPS as a minor issue. Given that AGT is a keyword this year, the paper will not be rejected just because of the topic; on the other hand, the decision will be made in comparison with other AGT submissions.

We hope the authors find the reviews helpful. Thanks for submitting to NeurIPS!